# Microbiome and epigenetic variation in wild fish with low genetic diversity

Ishrat Z. Anka [1,2], Tamsyn M. Uren Webster [1], Waldir M. Berbel-Filho [3,7], Matthew Hitchings [4], Benjamin Overland[1], Sarah Weller [1], Carlos Garcia de Leaniz [1,5] & Sofia Consuegra [1,6] ✉

Non-genetic sources of phenotypic variation, such as the epigenome and the microbiome, could be important contributors to adaptive variation for species with low genetic diversity. However, little is known about the complex interaction between these factors and the genetic diversity of the host, particularly in wild populations. Here, we examine the skin microbiome composition of two closely-related mangrove killifish species with different mating systems (self-fertilising and outcrossing) under sympatric and allopatric conditions. This allows us to partition the influence of the genotype and the environment on their microbiome and (previously described) epigenetic profiles. We find the diversity and community composition of the skin microbiome are strongly shaped by the environment and, to a lesser extent, by species-specific influences. Heterozygosity and microbiome alpha diversity, but not epigenetic variation, are associated with the fluctuating asymmetry of traits related to performance (vision) and behaviour (aggression). Our study identifies that a proportion of the epigenetic diversity and microbiome differentiation is unrelated to genetic variation, and we find evidence for an associative relationship between microbiome and epigenetic diversity in these wild populations. This suggests that both mechanisms could potentially contribute to variation in species with low genetic diversity.

Species with low genetic diversity have limited capacity for genetic-based adaptation under environmental change and have a higher risk of extinction[1], yet some can persist over many generations, suggesting that non-genetic sources of phenotypic variation (such as epigenetics or the microbiome) could also be involved in their adaptation to change[2,3]. Epigenetic modifications (i.e., DNA methylation, histone modifications, non-coding RNAs) modulate changes in gene expression that can occur in response to environmental variation but do not involve changes in DNA sequence[4,5], are mitotically and/or meiotically heritable and result in phenotypically plastic responses within

genotypes[6,7]. DNA methylation plays a role on the regulation of biological processes, such as cell differentiation and genomic imprinting, and can be affected by selection[8]. Many plants and animals display high levels of genome-wide DNA methylation[9-11] despite having low heterozygosity, suggesting that epigenetic variation might compensate for low genetic diversity and/or asexual reproduction[7,12,13], particularly when the variation is not under genetic control[14,15]. Like the epigenome, the microbiome, can increase host fitness by increasing phenotypic variation and the ability to respond to wider selective pressures, but also by buffering the host against environmental perturbations[16].

[1]Department of Biosciences, Centre for Sustainable Aquatic Research, Swansea University, Swansea, Wales SA2 8PP, UK. [2]Department of Aquaculture, Chattogram Veterinary and Animal Sciences University, Chattogram 4225, Bangladesh. [3]Department of Biology, University of Oklahoma, Norman, OK 73019, USA. [4]Institute of Life Science, Swansea University, Swansea, Wales SA2 8PP, UK. [5]Marine Research Centre (CIM-UVIGO), Universidade de Vigo, Vigo, Spain. [6]Grupo de Biotecnología Acuática, Departamento de Biotecnología y Acuicultura, Instituto de Investigacións Mariñas, IIM-CSIC Vigo, Spain. [7]Present address: Department of Biology, University of West Florida, Pensacola, FL, USA. ✉e-mail: s.consuegra@swansea.ac.uk

The interaction between the microbiome and the host genome can result in changes in gene expression without modifying the underlying DNA sequence, is strongly influenced by the environment, and can respond to selective pressures[17], therefore, could be considered an additional epigenetic mechanism of the host[18].

Both the microbiome and epigenome can influence host gene expression, and it is likely that there is a degree of interaction between them, but current understanding of the cross-talk between the microbiome and the genome and epigenome of the host, and their potential contribution to host plasticity, is still in its infancy[19]. In mammals, the host-gut microbiome interaction seems to be primarily mediated by microbiota-produced metabolites, such as short chain fatty acids (SCFAs), that modify the epigenome of gastrointestinal host cells through DNA methylation and histone acetylation, thereby altering the host cells' function[20]. Thus, changes in the microbiota composition or diversity can alter the production of metabolites that regulate host DNA and histone modifications[21]. Microbiome composition and function are influenced by the environment and by intrinsic host factors such as age, sex, immunocompetence and genotype, although their relative influence varies[22]. Host genetics tends to play a relatively small part in microbiome composition and involves few genes[23], influencing some tissues more than others[24,25]. However, population bottlenecks can reduce the diversity of the host and its microbiome, decreasing host fitness and its evolutionary response to stress[26], which makes the study of the interaction between the host and its microbiome very relevant for conservation[27].

The fish microbiome consists of a diverse community of bacteria, viruses, eukaryotes and protists associated to mucosal tissues in the gut[28], skin[29] and gills[30]. Its composition differs between organs, all of which have specialised microbiota[31]. The skin microbiota is strongly influenced by environmental factors, including water chemistry and bacterioplankton[32]. Yet, the microbiome of the fish skin is distinct from that of the surrounding water, and although it lacks a set of core taxa, it is mostly dominated by Proteobacteria[33]. The composition of the fish gut microbiome varies between fish genotypes, as does that of the skin and gills'[34,35], but the evidence for phylosymbiosis (higher intraspecific than interspecific similarity in the structuring of the microbial communities) is generally weak[36,37]. For example, host phylogenetics (but also diet) influence the skin microbiome composition of coral reef fishes[29] but not of Amazonian fishes[32], and an analysis of teleosts and elasmobranchs only found a consistent phylosymbiotic pattern in the latter[38]. Given that the fish skin mucus and its microbiome constitute the first barrier against infection[39], the influence of host genotype and the environment on its composition are likely to be important for the persistence of natural populations, potentially through epigenetic modulation. However, the influence of the environment and host genetics on skin microbiome and its relationship with host epigenetics has not yet been explored, and is particularly challenging in wild populations, where the ability to control for environmental conditions and genetic background is very limited. Studying closely related taxa under sympatric and allopatric conditions could help overcome this limitation, as this would allow us to account for environmental influences (shared under sympatry) and to examine genotype by environment interactions with microbial communities under natural conditions[40].

To examine the impact of environmental and genetic variation on the microbiome of species with low genetic diversity, as well as the association with host epigenetics and fluctuating asymmetry (a proxy for phenotypic fitness), we compared the skin microbiome composition of two closely related killifish species, *Kryptolebias ocellatus* and *Kryptolebias hermaphroditus*, with different mating systems (outcrossing and self-fertilisation respectively), that result in varying levels of genetic variation. We sampled locations where both species coexisted (sympatry) and where only one of them was present (allopatry), to control for environmental conditions, and assessed (a) the relative

effects of the environment, species and host genetic variation on the skin microbiome diversity and community structuring and (b) the relationship between microbiome diversity and host genetics, epigenetics (DNA methylation patterns) and fluctuating asymmetry.

## Results

Skin swabs were collected from two closely related mangrove killifish *Kryptolebias hermaphroditus* (*n* = 22; mean standard length = 28.27 SD = 4.90) and *Kryptolebias ocellatus* (*n* = 20; mean standard length = 28.80 SD = 7.17; Supplementary Data 1), from six sites in Brazil, two sites where both species coexist in sympatry (Guaratiba and Fundão; GUA and FUN), two sites only inhabited by *K. ocellatus* (Florianópolis and São Francisco do Sul; FLO and SFR) and two sites only inhabited by *K. hermaphroditus* (Picinguaba and Aracruz; PIC and ARA) as described in[41,42] (Fig. 1). *K. hermaphroditus* is one of only two known self-fertilising hermaphrodites in vertebrates[43] and its populations consists mainly of self-fertilising hermaphrodites with males at very low frequencies[44]. Outcrossing rarely occurs between *K. hermaprhoditus* males and hermaphrodites, which are typically inbred with very high homozygosity levels[45]. In contrast, *K. ocellatus* populations consist of males and hermaphrodites in approximately equal ratio and only reproduce via outcrossing[41].

### Differences in microbial composition between locations and fish species

The relative abundances of bacterial families differed between sampling locations, as shown by amplicon sequencing of the 16 S rRNA region. In sympatry, *Phycisphaeraceae, Arcobacteraceae, Sulfurovaceae* were the three most abundant families for both species at location GUA, while *Arcobacteraceae, Moraxellaceae* and *Vibrionaceae* were the most abundant at FUN. Under allopatry, the five most abundant families in *K. ocelatus* were *Phycisphaeraceae, Rhodobacteraceae, Desulfosarcinaceae, Pirellulaceae* and *Anaerolineaceae* at SFR and *Wohlfahrtiimonadaceae, Sphingomonadaceae, Rhodobacteraceae, Anaerolineaceae* and *Flavobacteriaceae* at FLO. For *K. hermaphroditus* the most abundant families were *Vibrionaceae, Pseudoalteromonadaceae, Comamonadaceae, Moraxellaceae* and *Arcobacteraceae* at PIC and *Rhodobacteraceae, Thermaceae, Solimonadaceae, Moraxellaceae* and *Vibrionaceae at* ARA (Fig. 2A).

The two species also displayed significant differences in ASV composition at the sympatric locations, with 46 ASVs being significantly different in abundance at GUA and 20 at FUN (Table 1). The highest differences at location GUA corresponded to *Sulfurovum* (represented by at least three different ASVs with different abundance in both fish species; *P* < 0.001), *Arcobacteraceae* (*P* < 0.001) and *Sulfurimonas* (*P* < 0.001), all more abundant in *K. ocellatus*. At FUN, the main differences corresponded to *Anoxybacillus* (*P* = 0.010), *Meiothermus* (*P* = 0.030) and *Intrasporangiaceae* (*P* = 0.008; more abundant in *K. hermaphroditus*) and *Desulfosarcina* (*P* = 0.010) and *Rhodobacteriaceae* (*P* = 0.025; more common in *K. ocellatus*) (Fig. 2B,C).

Some ASVs were significantly associated with certain species and locations. In sympatry, 13 and 26 indicator taxa were identified at GUA and FUN, respectively (Supplementary Data 2), whereas 3 and 19 were identified for *K. hermaphroditus* and *K. ocellatus* (Supplementary Data 3). These results suggest that location played a more important role than species in microbiome differentiation, and that *K. ocellatus* (the outcrossing species) displayed a higher proportion of unique taxa than *K. hermaphroditus* (selfing species). A similar pattern was found between species when all the locations were pooled, with 61 indicative taxa for *K. ocellatus* compared to 25 for *K. hermaphroditus*. Indicative taxa for locations displayed a higher relative abundance than those from different species (Supplementary Data 2,3). Indicative species-specific ASVs included different ribotypes of *Sulfurovum*, which

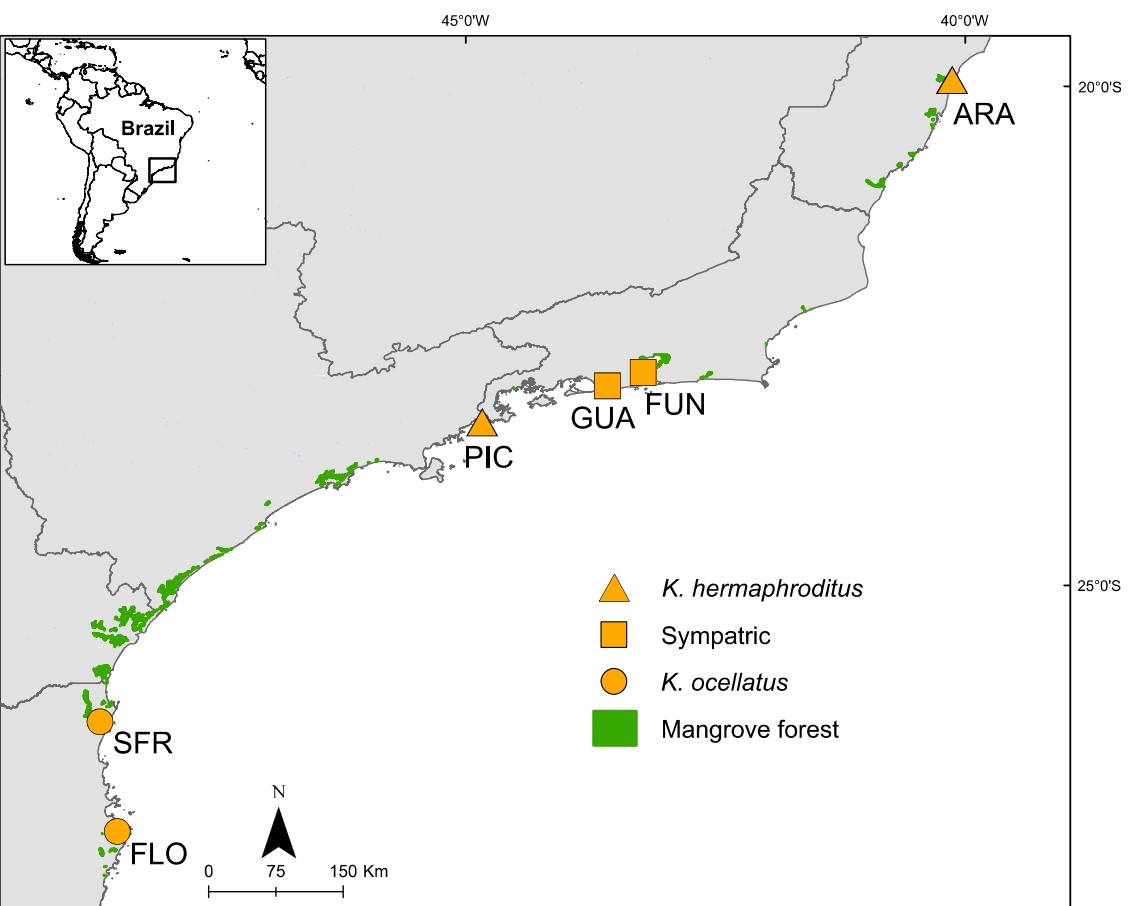

**Fig. 1 | Sampling locations for *Kryptolebias ocellatus* and *Kryptolebias hermaphroditus*.** Includes locations with species in allopatry: FLO, SFR, PIC, and ARA and sympatry: FUN and GUA. GUA = Guaratiba, FUN = Fundão, Florianópolis = FLO, São Francisco do Sul = SFR, Picinguaba = PIC, Aracruz = ARA.

oxidises sulphur and thiosulfate and is found in the gut and gill microbiome of marine invertebrates (like sea cucumber and snails) where it could be providing detoxification and nutritional intake for the host[46]. Indicative ASVs related to location included *Phycisphaeraceae*, a member of the scarcely studied class Phycisphaerae, common in the marine environment[47,48] and the microbiome of freshwater fish living in anoxic conditions[49], such as those found in mangrove killifish habitats[50] (Supplementary Data 4).

The predicted community metagenomic profiling based on MetaCyc pathway data for prokaryotes identified 17 and 2 functional traits that differed between locations within *K. ocellatus* and *K. hermaphroditus* populations, respectively, when all locations were considered. In contrast, no functional differences were identified between species in the shared locations, indicating that species-indicative ASVs in those habitats had likely redundant functions. Functional traits enriched in the different locations included the mevalonate pathway I and isoprene biosynthesis II (engineered) for *K. ocellatus* and L-arginine degradation II (AST pathway) and cob(II)yrinate a,c-diamide biosynthesis II (late cobalt incorporation) for *K. hermaphroditus* (Supplementary Fig. 1).

### Species and sampling location influence microbiome alpha and beta diversity

We assessed the influence of species, sampling location and fish size on measures of alpha diversity (Shannon diversity, Chao1 richness, Simpson's evenness and Faith's phylogenetic diversity) (Supplementary Data 1). Body size (standard length) did not have a significant effect on Chao1 (F = 0.486, P = 0.49), Faith (F = 0.022, P = 0.88) or Simpson (F = 2.481, P = 0.12) diversity measures. Species,

but not location, had a significant effect on Faith PD (phylogenetic diversity) and Chao1 (richness) measures of skin microbiome diversity (Faith: F = 0.005, P = 0.006); Chao: F = 6.771, P = 0.013), with *K. ocellatus* displaying higher diversity than *K. hermaphroditus* in both cases (Faith KOce mean=31.32 SD = 13.10; KHer mean = 17.86 SD = 8.81; Chao: KOce mean = 422.69 SD = 263.91, KHer mean = 213.68 SD = 130.69). Only location influenced Simpson's evenness, which measures species' dominance (Species: F = 0.301; P = 0.58; Location F = 8.039, P < 0.001), which was also higher in *K. ocelatus* than in *K. herpmaphroditus* (KOce mean = 0.23 SD = 0.15, KHer mean =0.19 SD = 0.13). Shannon diversity (which takes abundance and evenness into account) was only significantly influenced by size (Species F = 0.440, P = 0.512; Location F = 1.586, P = 0.191; Size F = 5.851, P = 0.021) (Fig. 3). Bootstrapping analyses, with 1000 dataset replicates, supported the results of these models (Supplementary Data 5, Supplementary Fig. 2). Multivariate analysis of community separation (PERMANOVA) indicated that species, location and their interaction had a significant effect both on Bray-Curtis dissimilarity and weighted UniFrac distance (Supplementary Data 6), with location explaining the highest percentage of the data in both cases (Bray-Curtis: Location $R^2 = 0.265$ P < 0.001; Species $R^2 = 0.037$ P = 0.002; Location*Species $R^2 = 0.004$ P = 0.004; UniFrac: Location $R^2 = 0.397$ P < 0.001; Species $R^2 = 0.039$ P = 0.026; Location*Species $R^2 = 0.039$ P = 0.028). Group visualisation by NDMS using Bray-Curtis distance revealed the influence of location and species in the structural diversity of skin microbiome, as both species were intermingled in the shared locations (FUN and GUA) but tended to group by species when originated in separate locations (Fig. 4a). NMDS structuring based on weighted UniFrac distance was less clear,

A)

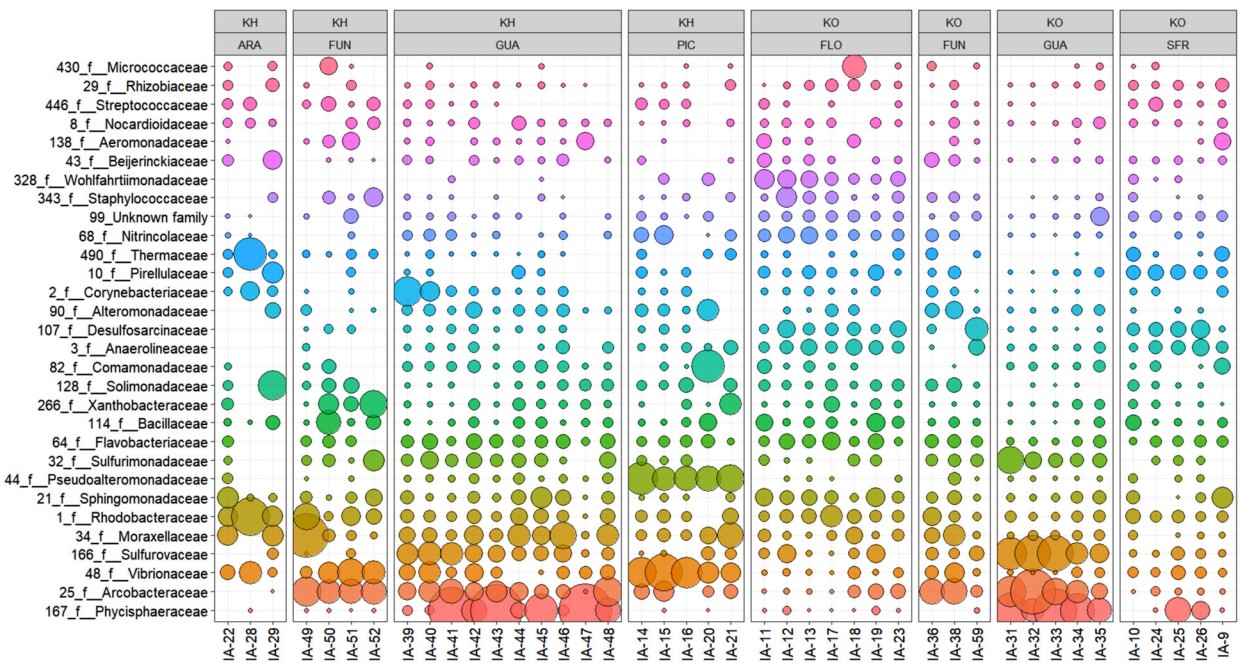

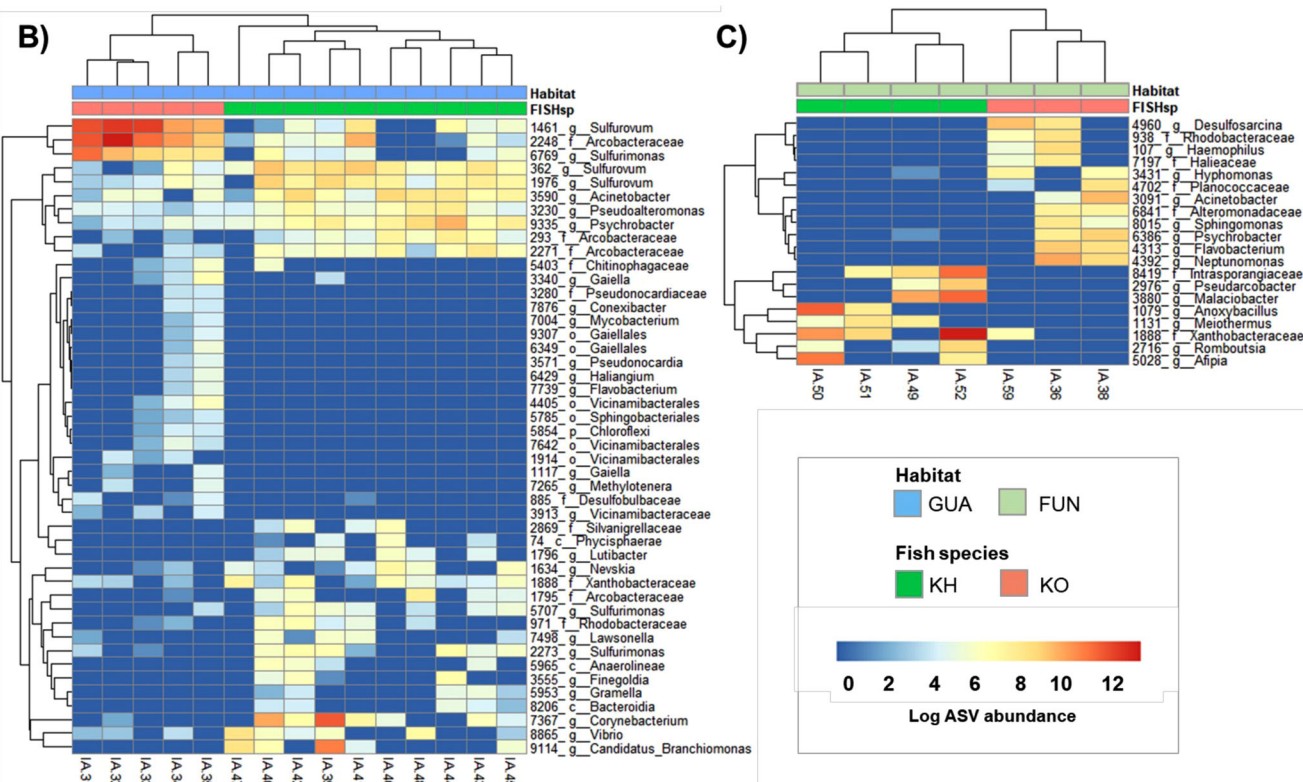

**Fig. 2 | Distribution of microbiome families across species and locations.**
**A** Most abundant 30 families based on 12,844 subsampled reads, separated by species and location (single or shared). Columns represent individual fish.

**B** Significant differences in ASV composition between species in sympatry in GUA sampling location and **C** in FUN. GUA = Guaratiba, FUN = Fundão, Florianópolis = FLO, São Francisco do Sul = SFR, Picinguaba = PIC, Aracruz = ARA.

apart from the samples from GUA that clustered together and were more distant from the rest (Fig. 4b).

When we considered the occurrence of both species in sympatry or allopatry, Chao1 diversity was significantly influenced by species and sympatry (Species: F = 10.819 P = 0.002; Sympatry: F = 4.575 P = 0.038),

and so was Faith PD (Species: F = 17.118 P = 0.002; Sympatry: F = 5.780 P = 0.02), unlike Shannon diversity (Species: F = 3.486 P = 0.069; Sympatry: F = 2.177 P = 0.148) or Simpson's evenness which was influenced only by sympatry (Species: F = 0.641 P = 0.428; Sympatry: F = 11.638 P = 0.002). Chao1 and Faith_PD diversities were higher in *K. ocellatus* than

in *K. hermaphroditus*, and higher for both species when they were in sympatry compared to allopatry, while the latter trend was the opposite for Simpson's evenness (Fig. 3, Supplementary Fig. 3).

## Individual genetic diversity influences microbiome diversity

To assess the influence of genetic variation on microbiome diversity, individual heterozygosity (He) was calculated at FLO, SFR, FUN and GUA locations previously genotyped for 5,477 SNPs[51] (Supplementary Data 1). *K. hermaphroditus* ($n = 14$) had an average individual heterozygosity of 0.04 (SD = 0.01) while *K. ocellatus'* average individual heterozygosity ($n = 14$) was 0.08 (SD = 0.01), the latter being higher than expected from

their respective mating systems, self-fertilising in *K. herpmaphroditus* and outcrossing in *K. ocellatus*. Comparisons between sympatric and non-sympatric populations indicated that species (F = 186.571 $P < 0.001$) and sympatry (shared or non-shared location; F = 28.101 $P < 0.001$) both influenced individual heterozygosity, with *K. ocellatus* displaying higher heterozygosity when coexisting with *K. hermaphroditus* than in locations where it lived in isolation (there was no Data available on He for *K. hermaphroditus* in isolation) (Supplementary Fig. 4).

Full models of microbiome alpha diversity included species, location, size and individual heterozygosity as predictors. Model checks carried out using the *performance* package indicated collinearity between species and heterozygosity, and species was removed from the model. Stepwise model selection using the *drop1* command indicated that individual heterozygosity (F = 6.192 $P = 0.020$) and location (F = 4.353 $P = 0.014$) significantly affected Chao1 richness and Faith phylogenetic diversity (heterozygosity F = 8.338 $P = 0.008$ and F = 6.573 $P = 0.022$). Fish size (F = 12.451 $P = 0.002$) and location (F = 6.191 $P = 0.003$) significantly influenced Shannon diversity and only location significantly influenced Simpson's evenness (F = 7.88 $P = 0.009$) (Fig. 5a–d). Non-parametric bootstrapping regressions based on 1000 repeats supported these results (Supplementary Data 4). Mantel tests between genetic (based on SNPs) and microbiome distance matrices, carried out using 10,000 permutations, indicated a weak but significant positive correlation between Euclidean genetic distance and weighted Unifrac microbiome dissimilarity (which considers microbiome phylogenetic distance; Mantel R = 0.155 $P = 0.047$) but no significant correlation with Bray-Curtis microbiome dissimilarity (Mantel R = −0.134 $P = 0.108$; Supplementary Fig. 5).

## Host microbiome and genetic differentiation are associated with DNA methylation

To assess the relationship between microbiome and host DNA methylation patterns, we used data on genetic (SNP-based) and

**Table 1 | Pairwise comparison of ASV abundances between localities for both mangrove killifish species in sympatric (GUA, FUN) and allopatric locations**

| Species | Locations | No. ASV differences |
|---|---|---|
| *K. hermaphroditus* | FUN v ARA | 46 |
| | FUN v GUA | 147 |
| | FUN v PIC | 79 |
| | GUA v ARA | 104 |
| | PIC v ARA | 60 |
| | PIC v GUA | 170 |
| *K. ocellatus* | FUN v FLO | 314 |
| | GUA v FLO | 321 |
| | SFR v FLO | 286 |
| | FUN v GUA | 147 |
| | FUN v SFR | 201 |
| | GUA v SFR | 347 |

*GUA* Guaratiba, *FUN* Fundão, *Florianópolis* FLO, *São Francisco do Sul* SFR, *Picinguaba* PIC, *Aracruz* ARA.

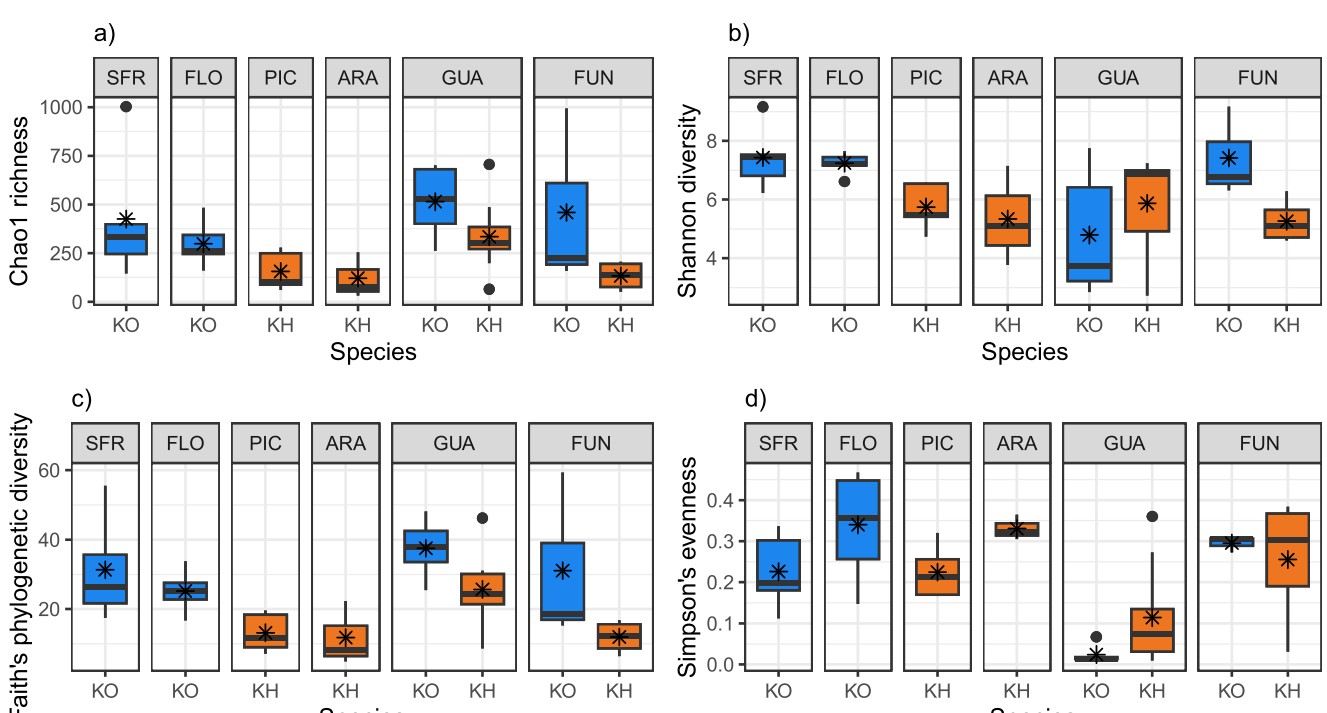

**Fig. 3 | Alpha diversity measures of the skin microbiome. a** Chao1 species richness, **b** Shannon diversity, **c** Faith's phylogenetic diversity and **d** Simpson's eveness. Including *Kryptolebias ocellatus* (KO) and *Kryptolebias hermaphroditus* (KH) from sympatric (GUA and FUN) and allopatric locations (SFR, FLO, PIC and ARA) locations. Number of biologically independent samples: SFR-KO $n = 5$, FLO-KO $n = 7$,

PIC-KH $n = 5$, ARA-KH $n = 3$, GUA-KH $n = 10$, GUA-KO $n = 5$, FUN-KH $n = 4$, FUN-KO $n = 3$. The black centre line denotes the median value, the coloured box contains the 25th to 75th percentiles of Dataset, the whiskers mark the 5th and 95th percentiles, and values beyond these upper and lower bounds are considered outliers, means are represented by asterisks. Source Data are provided as a Source Data file.

epigenomic (DNA methylation) pairwise distances between individuals, previously estimated in[42] for 18 fish occurring in sympatry (Supplementary Data 7). Multiple regression analyses (MRM) carried out using 10,000 permutations indicated a significant relationship between epigenetic, genetic and microbiome distance matrices ($R^2 = 0.435$ $P = 0.001$), with methylation dissimilarity being significantly and positively correlated to both genetic Euclidean ($P = 0.001$) and microbiome Bray-Curtis ($P = 0.001$) distance (Fig. 6). When the analysis was run with weighted Unifrac distance instead of Bray-Curtis for the microbiome dissimilarity, the relationship was still significant ($R^2 = 0.131$ $P = 0.001$) but only the Euclidean genetic distance was significantly correlated to the methylation Bray-Curtis dissimilarity (Euclidean: $P = 0.001$; Unifrac $P = 0.921$). As a measure of individual epigenetic diversity, we also estimated the coefficient of variation (CV) of the counts per million of 64,152 methylated sites[42]. Simpson's evenness index was the only alpha diversity metric significantly associated with methylation CV, with the best model including just Simpson eveness ($F = 25.12$ $P < 0.001$).

### Fluctuating asymmetry correlates with host microbiome and genetic differentiation

Fluctuating asymmetry (FA; i.e., the random deviation from symmetry in bilateral organisms)[52] is a phenotypic indicator of developmental instability often associated with environmental or genetic stress, although its relationship with fitness is unclear[53]. We measured FA in three traits: pupil diameter and distance from the eye to the snout, previously shown to exhibit FA in fish[54] and area of the caudal ocellus, a dark spot present in *Kryptolebias hermaphrodites* and secondary males which has been associated with aggressive behaviour[55]. Only two traits (ocellus area and pupil diameter) displayed FA and were retained for the analysis, while the eye-snout distance displayed antisymmetry. Total FA was higher in *K. ocellatus* than in *K. hermaphroditus* (t = −2.0886, df = 19, $P = 0.05$) (Supplementary Fig. 6a–c) and positively correlated with Faith phylogenetic distance (R = 0.55, Pearson $P = 0.010$, Permutation-based

$P = 0.006$), Chao1 diversity (R = 0.51, Pearson $P = 0.019$, Permutation-based $P = 0.020$) and individual heterozygosity (R = 0.45, Pearson $P = 0.039$, Permutation-based $P = 0.044$) (Fig. 7) but not with methylation coefficient of variation (R = −0.24, Pearson $P = 0.422$, Permutation-based $P = 0.414$) (Supplementary Fig. 6d).

## Discussion

Current understanding of how the microbiome and the host (epi)genome contribute to host phenotypic plasticity is still limited, despite their potentially important influence on adaptation[56], particularly in populations with low genetic diversity. We assessed the potential association of microbiome and epigenetic variation of two closely related fish species with contrasting mating systems and variable levels of genetic diversity, living in sympatry and in allopatry, to analyse their potential contribution to variation in wild populations with low genetic diversity.

### Species and location as drivers of skin microbiome composition and diversity

The skin microbiome of the mangrove killifishes was dominated by Proteobacteria at the phylum level, followed by Campilobacterota, Planctomycetota, Actinobacteriota, Bacteroidota and Firmicutes, all of these commonly present in the fish microbiota[57,58]. Differences between sympatric and allopatric populations highlighted the influence of the environment on microbiome composition and diversity, more pronounced in *K. ocellatus* than in *K. hermaphroditus*, but differences between both species in sympatry also indicated a species-specific effect on the skin microbiome. Fish species and location affected alpha diversity in different ways. While species seemed to influence more microbial ASV richness and phylogenetic diversity, the location had a stronger influence on ASV evenness and dominant community members. The interaction between location and species was also observed in the microbiome population structuring, measured by beta diversity, but the environment explained more of the differentiation, as for other fish species like Atlantic salmon[59]. A similar pattern is found in the gut microbiome of mangrove crabs, that display species-specific microbiome metagenomic profiles but also the influence of the environment, resulting in a large amount of OTUs shared between species[60]. In general, the fish skin supports a very diverse microbiome community, different from the surrounding water and variable at different levels, from species to individuals and organs[61] and the influence of species-specific factors and location on the microbiome seems very variable among fish groups[32,37,62], probably resulting from the large fish diversity and long evolutionary history[63]. We found strong differences in ASV composition but also functional redundancy between the microbiome of both species living in sympatry. Functional redundancy of the fish microbial community occurring at different scales (e.g., local communities or habitats) acts as a spatial ecological insurance within ecosystems, ensuring the maintenance of key ecological processes within and across habitats[64]. Thus, the functional redundancy observed between both species in sympatry could also be a reflection of their different colonisation histories, with *K. hermaphroditus* having only recently colonised those locations[41,65].

### Interactions between microbiome, host genetic diversity and epigenetics

The role of genetics in determining the microbiome composition has been mainly discussed in terms of species specificity, in fish and other taxa[66,67], evidenced by the concordance found between host phylogenies and microbiome assemblages[68]. Population genetic divergence also influences microbiome differentiation in fish[69]. The populations we analysed displayed a natural gradient of individual heterozygosity, which correlated positively with both microbial phylogenetic diversity and richness. *K. ocellatus* (the outcrossing and genetically more variable species), displayed more microbiome differentiation and diversity within and between locations than its self-fertilising counterpart,

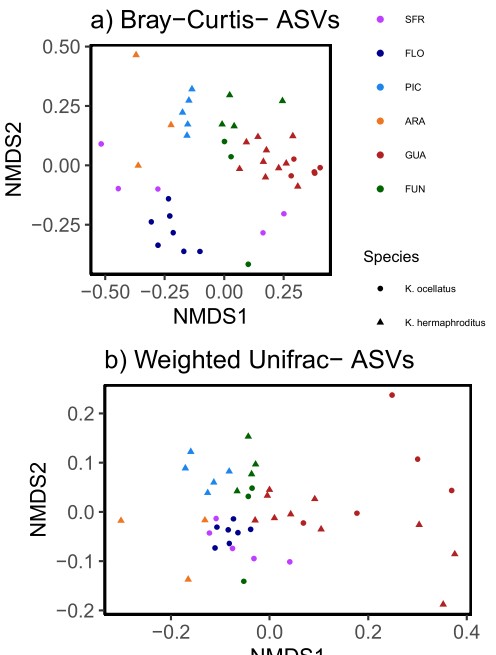

**Fig. 4 | Beta diversity measures of skin microbiome.** Non-metric multidimensional scaling (NMDS) ordination of the microbial skin community of *Kryptolebias ocellatus* (circles, n = 20) and *Kryptolebias hermaphroditus* (triangles, n = 22) from shared (GUA and FUN) and separate (SFR, FLO, PIC and ARA) locations based on (**a**) Bray-Curtis distance and (**b**) weighed UniFrac distance.

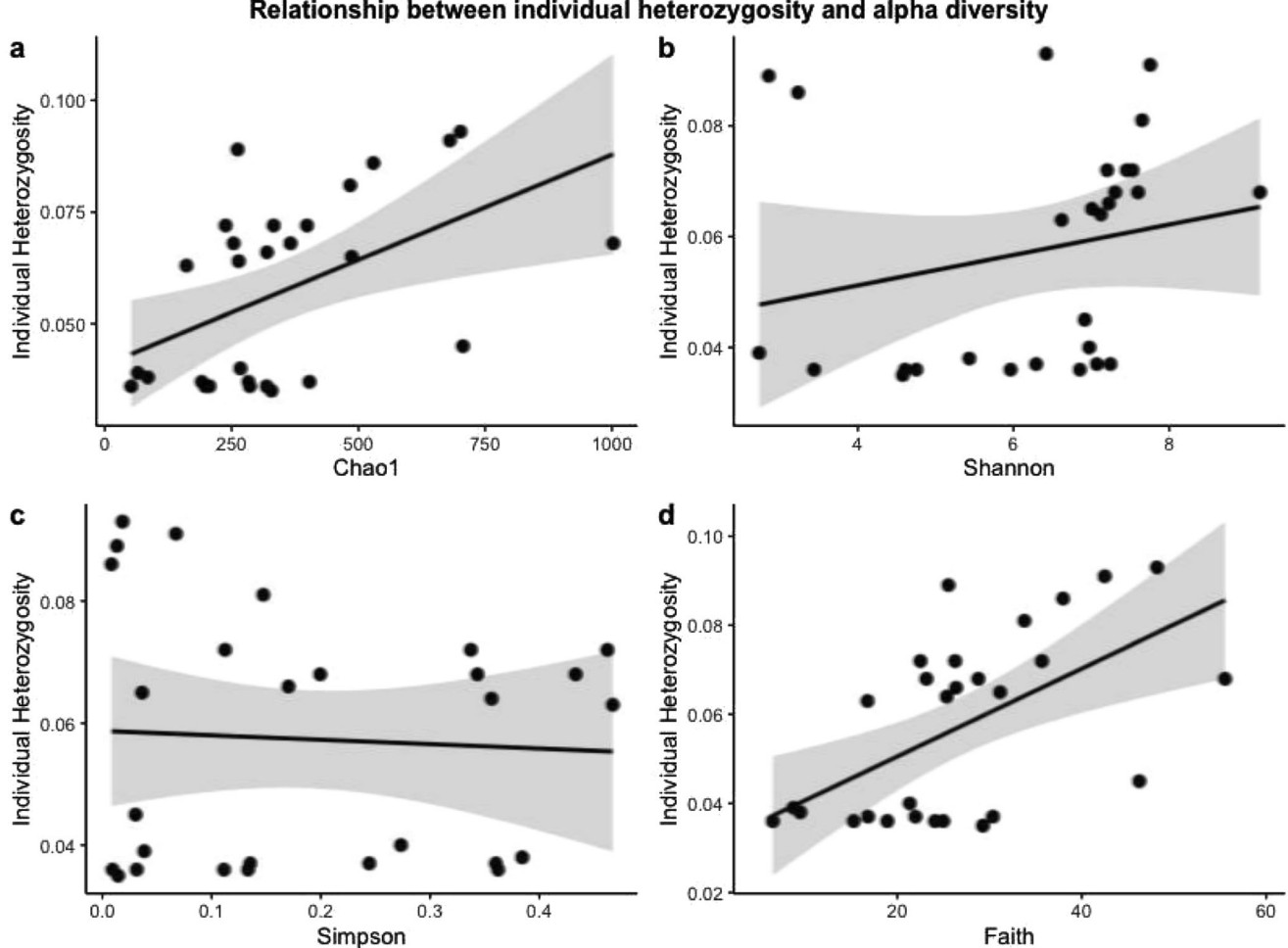

**Fig. 5 | Relationship between individual heterozygosity and estimates of skin microbiome alpha diversity.** Alpha diversity measurements include (**a**) Chao1, (**b**) Shannon index, (**c**) Simpson's evenness index and (**d**) Faith phylogenetic distance for *Kryptolebias ocellatus* and *Kryptolebias hermaphroditus* (pooled, *n* = 28) in the sympatric (FUN, GUA) and allopatric locations (SFR, FLO). Grey bands represent 95% confidence intervals. Data were analysed using linear models: lm(alpha diversity ~ Species + SL + Location + H.indiv). Individual heterozygosity significantly influenced Chao1 richness (F = 6.192, P = 0.020) and Faith phylogenetic diversity F = 8.338 P = 0.008). Non-parametric bootstrapping regressions based on 1000 repeats supported these results (Supplementary Data 5). Source Data are provided as a Source Data file.

reflecting the strong relationship between microbiome and genetic diversity. Given the influence of environment and species on microbiome composition, the observed relationship between heterozygosity and the microbiome could also reflect co-variation between species and genetic diversity, driven by the selective pressures imposed by environmental heterogeneity on population sizes and genetic diversity of both species[70,71]. This result highlights the importance of considering microbiome diversity for conservation[72], particularly in the face of rapid environmental change, which also affects the microbiome. The key role of both microbiome and genetic diversity in host fitness, and the implications that low genetic diversity and inbreeding have in reducing host immunocompetence[73], mean that reduced genetic and microbial host diversity could interact to reduce host resilience to environmental change[26]. However, despite the lower genetic and microbial diversity of the self-fertilising *K. hermaphroditus*, these populations are stable or even expanding across their range[41], suggesting that alternative sources of plasticity could also play a role in their adaptation to environmental change.

It has recently been suggested that the microbiome, which is influenced by the host genetics and environmental selective pressures[17], could be considered as an additional epigenetic mechanism of the host[19], and that the holobiont (host and microbiome with their respective genomes)[17] could be the target of selection. Our results indicate an association between the host genetics and skin microbiome with the host epigenetics (DNA methylation). Epigenetic pairwise distance between individuals was positively correlated with microbiome differentiation and genetic dissimilarity. In addition, fish with higher coefficient of variation in DNA methylation (used here as a rough estimation of epigenetic individual variability) displayed higher alpha diversity (evenness) in their skin microbiome. At least part of this association could reflect the close relationship between epigenetic and genetic diversity, which we have previously observed in the sister species of *K. hermaphroditus*, *K. marmoratus*, also self-fertilising, reared under different environmental conditions[74]. Our previous Data also indicated that an interaction between parasite loads (including gill bacterial cysts) and genetic diversity influenced DNA methylation patterns in wild *K. hermaphroditus* populations[75]. However, variation in epigenetic diversity (CV) was not explained by heterozygosity and, of all measurements of alpha diversity, only Simpson's evenness (unrelated to heterozygosity) had an influence on its distribution. This suggests that a proportion of epigenetic diversity is not directly related to host genetic diversity. In the closely related *K. marmoratus*, we had previously found that there was a small proportion of epigenetic diversity associated with the rearing environment, but not with the

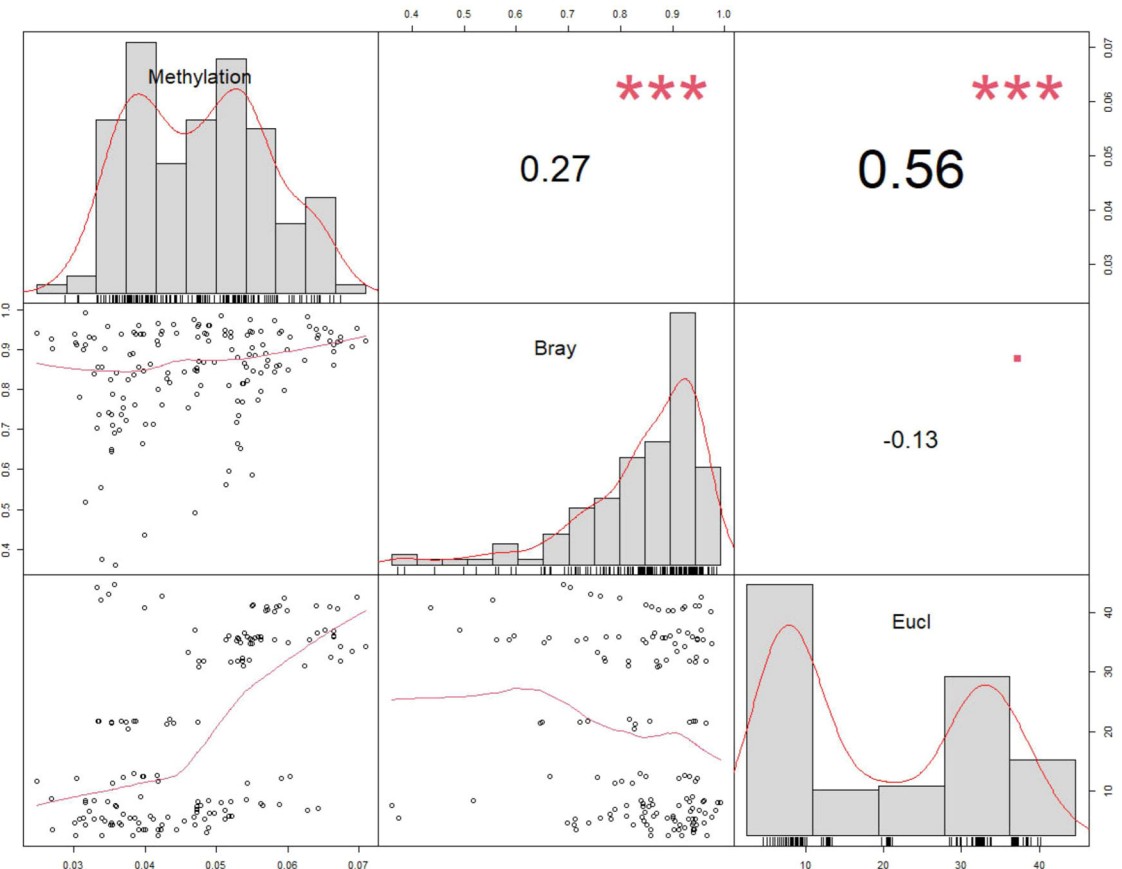

**Fig. 6 | Relationship between epigenetic, genetic and microbiome pairwise distances.** Epigenetic = DNA Methylation, Bray-Curtis; genetic = SNPs, Euclidean; microbiome = Bray-Curtis. Pairwise distances between 18 individuals (14 *Kryptolebias hermaphroditus* and 4 *K. ocellatus*) in sympatry (GUA and FUN sampling locations), including variables distribution, value of the correlation and significance based on Pearson tests (see main text for multiple regression analysis on distance matrices, MRM). Source Data are provided as a Source Data file.

genotype, that might be maintained in the next generation[74,76]. Stochastic and pure epigenetic epimutations (driven by the environment) which can persist over generations[77] have been suggested as a potential bet-hedging strategy, particularly relevant for populations with low genetic diversity[15]. Microbiome dissimilarity (both Bray-Curtis and Unifrac differentiation) was also uncorrelated with genetic differentiation, indicating that the more genetically similar fish (in this case those originating from *K. hermaphroditus* self-fertilisation) did not share a more similar microbiome, which instead was more related to the sampling location.

As with most field studies, our Data is observational, and this makes it difficult to determine the direction or causality of the observed relationships or draw inferences about the adaptive response. We attempted to overcome this challenge by measuring fluctuating asymmetry in traits related to fish performance. Fluctuating asymmetry in phenotypic traits is often used as a proxy for fitness[78], based on its relationship with heterozygosity, stress and inbreeding, but the significance and strength of this association is inconsistent among studies[53]. We identified a positive association between FA and heterozygosity, which does not fit the 'heterozygosity theory', according to which more heterozygous individuals should be developmentally more stable than their more homozygous counterparts, due a higher metabolic efficiency[79]. This relationship would, however, fit the 'genomic coadaptation theory', according to which developmental stability can decline if coadapted gene complexes are disrupted, for example, by gene flow or introgression[79]. *K. hermaphroditus* is naturally highly inbred and displays no evidence of inbreeding depression[80], thus its self-fertilising reproduction could

result in a purge of deleterious alleles and in more balanced coadapted gene complexes, which would increase developmental stability and therefore decrease asymmetry[81]. Despite their different mating systems, both species are able to hybridise when occurring in sympatry, and we previously found evidence of backcrosses of the hybrids with *K. ocellatus*[51]. Although none of the fish analysed here were classified as hybrids, the hybridisation history among mangrove killifishes[82] means that there is potential for gene flow between both species, with introgression likely affecting mostly *K. ocellatus*. Interspecific hybridisation can result in outbreeding depression and breakdown of coadapted gene complexes, which could increase developmental instability and fluctuating asymmetry, at least in some traits, potentially affecting several generations[83]. The asymmetric direction of hybridisation could also be a source of genetic stress reducing developmental stability in *K. ocellatus*. In addition to heterozygosity, both Faith and Chao1 measures of microbiome diversity, but not epigenetic diversity, were associated with FA.

Here, we show that both environment and species play a role in shaping the microbiome diversity and community composition of the mangrove killifish. Genetic, epigenetic and microbiome diversity display a complex relationship, where heterozygosity and microbiome alpha diversity, but not epigenetic variation, are associated with the fluctuating asymmetry of traits related to fish performance (vision) and behaviour (aggression). We also identify epigenetic diversity and microbiome differentiation that are independent of host heterozygosity or genetic differentiation but associated with each other. We cannot ascertain whether this association is due to the production of microbial metabolites regulating the epigenome (as in mammals[20,21]),

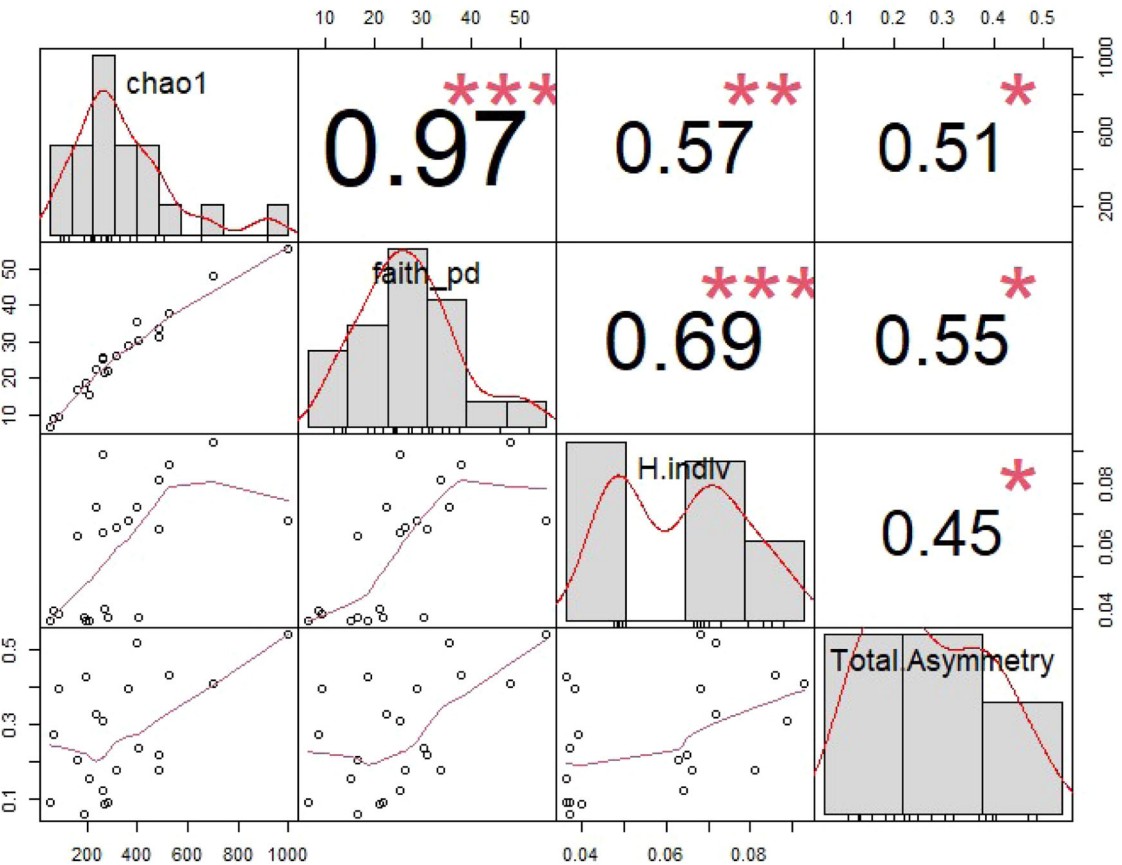

**Fig. 7 | Relationship between total fluctuating asymmetry, genetic diversity (individual heterozygosity) and microbiome alpha diversity.** Based on 21 individuals (10 *Kryptolebias hermaphroditus* and 11 *K. ocellatus*), including variables distribution, value of the correlation and significance based on Pearson tests (see main text for additional probabilities based on 1000 permutations). Source Data are provided as a Source Data file.

to the influence of the host epigenome on the microbiome, or co-variation in response to environmental pressures[71]. Yet, irrespective of its origin, the proportion of epigenetic and microbiome diversity unrelated to host genetics could provide an additional source of variation, potentially very important for fish with low genetic diversity.

## Methods

### Species selection and sampling

Sampling was carried out under license ICMBio/SISBIO 57145-1/2017 (which included an exportation permit) and approved by Swansea University Ethics Committee reference SU-Ethics-Student-250717/245. The sampling took place in 2017, before Brazil ratified the Nagoya protocol in 2021, however we ensured to follow Brazilian national laws, get prior informed consent from the authorities to carry out the sapling through our local partners and shared benefits fairly and equitably.

*Kryptolebias hermaphroditus* and *Kryptolebias ocellatus*, two closely related mangrove killifish, were sampled from six sites in south and southeast Brazil, two sites where both species coexisted in sympatry (Guaratiba and Fundão; GUA and FUN), two sites only inhabited by *K. ocellatus* (Florianópolis and São Francisco do Sul; FLO and SFR) and two sites only inhabited by *K. hermaphroditus* (Picinguaba and Aracruz; PIC and ARA) as described in[41,42] (Fig. 1). *K. hermaphroditus* is one of the only two known self-fertilising hermaphrodites in vertebrates[43] and its populations consists mainly of self-fertilising hermaphrodites with males at very low frequencies[44]. Outcrossing rarely occurs between *K. hermaprhoditus* males and hermaphrodites, which are typically inbred with very high homozygosity levels[45]. In contrast, *K. ocellatus* populations consist of males and hermaphrodites

in approximately equal ratio and only reproduce via outcrossing[41]. Skin swabs of the left flank of the fish (between the operculum and caudal fin) were collected for this study from forty-two mangrove killifish: 8 *K. ocellatus* and 14 *K. hermaprhoditus* from sympatric locations, and 12 *K. ocellatus* and *8 K. hermaprhoditus* from allopatric locations (Table 2; Fig. 1). The swabs were stored in molecular grade ethanol at −80 °C until analysis. Fish standard length (SL, mm) was measured in the field (FUN and GUA) or from ethanol-stored specimens using the following empirical relationship SL fresh = 0.9246 * SL ethanol + 3.012 ($R^2$ = 0.96). We used our previous Data on single nucleotide polymorphism (SNP) diversity and DNA methylation patterns of *K. ocellatus* and *K. hermaprhoditus* sampled in sympatry at GUA and FUN locations[42] to assess the potential relationship between individual genetic and microbiome diversities, as well as the potential relationship between the microbiome community structure and epigenetic differentiation.

### DNA extraction, library preparation, and sequencing

The DNeasy PowerLyzer PowerSoil Kit (QIAGEN) was used to extract the microbial DNA from the skin swab samples[59]. Amplification of the 16 S rRNA-V4 region[84] was performed using the 515F-806R primers[85] with updated sequences 515 F:GTGCCAGCMGCCGCGGTAA[86] and 806 R:GGACTACHVGGGTWTCTAAT[87]. PCR_1 consisted of a total volume of 22.5 μL incorporating 12.5 μL of Platinum™ II Hot-Start PCR Master Mix (2X) (Thermo Fisher Scientific), 0.5 μL of Forward (FP) and Reverse (RP) primers (10 uM), 9 μL of Ultra-pure water (UPW) and 2.5 μL of DNA. The PCR began with a 3 min denaturation step at 95°C followed by 28 cycles of 95°C for 30 seconds, 55°C for 30 seconds and 72°C for 30 seconds, then a final elongation step at 72 °C for 5 minutes. During PCR_2, indexing with Nextera ® XT Index Kit v2 (Illumina, Inc.,

**Table 2 | Species and sampling locations**

| Species | Species code | Location code | Sampling location | Sample size n = 42 |
|---|---|---|---|---|
| *Kryptolebias ocellatus* | KO | FLO | Poço das Pedras, estuário do rio Ratones, Florianópolis, SC | 7 |
| *Kryptolebias ocellatus* | KO | SFR | Manguezal no canal do Linguado, São Francisco do Sul, SC | 5 |
| *Kryptolebias hermaphroditus* | KH | PIC | Manguezal do rio da Fazenda, P. E. S. M. Picinguaba, Picinguaba, SP | 5 |
| *Kryptolebias hermaphroditus* | KH | ARA | Alagado na praia de Coqueiral, Aracruz, ES | 3 |
| *Kryptolebias ocellatus* | KO | GUA | Manguezal do rio Piracao, Guaratiba, RJ | 5 |
| *Kryptolebias hermaphroditus* | KH | GUA | Manguezal do rio Piracao, Guaratiba, RJ | 10 |
| *Kryptolebias ocellatus* | KO | FUN | Manguezal da Ilha do Fundao, Rio de Janeiro, RJ | 3 |
| *Kryptolebias hermaphroditus* | KH | FUN | Manguezal da Ilha do Fundao, Rio de Janeiro, RJ | 4 |

San Diego, CA, 92122 United States) was performed. PCR_2 was made with a total volume of 27.5 μL per sample, containing 2.5 μL of PCR_1 product, 1.25 μL of each index, 12.5 μL of Platinum™ taq and 10 μL of UPW. The reaction conditions were as above but with 12 cycles. Final PCR products were pooled based on agarose gel band intensity and cleaned using AMPure XP beads (Beckman Coulter Genomics, Brea, CA, United States). Final library quantification was performed using qPCR (NEB Illumina quantification kit), prior to sequencing on a MiSeq Illumina platform (300 bp, paired end). Blanks were sequenced alongside the samples and yielded 206 reads.

## Bioinformatics analysis

Sequence analysis was performed as described in[88] using Qiime2 (version: qiime2-2022.2)[89]. Briefly, based on quality filtering, DADA2[90] was used to trim leading primers and truncate forward (220 bp) and reverse (180 bp) reads, denoise and merge reads, remove chimeras and assign amplicon sequence variants (ASVs). Filtering of mitochondrial, chloroplast and unclassified reads was carried out before sub-sampling a total of 12,844 reads (10.20% ASVs retained) and further removal of ASVs with total abundance of less than 2 across all samples leaving a total of 9,598 ASVs. Classification was then performed using the Silva reference taxonomy (v138)[91]. QIIME2[89] was used to estimate alpha diversity (Chao1 richness, Shannon diversity, Faith's phylogenetic diversity and Simpson's evenness). Beta diversity between sample pairs was calculated using Bray-Curtis and weighted UniFrac distances.

Statistical differences in ASV abundance were examined using DeSeq2[92]. Individual DeSeq models were constructed to identify differentially abundant ASVs occurring between fish species (KO and KH) present in the same environment (FUN and GUA), and, for each fish species separately, to assess the effect of different locations on ASV abundance. Low coverage ASVs were independently filtered within DeSeq2, and default settings were applied for outlier detection and moderation of ASV dispersion. ASV abundance was considered significantly different at FDR < 0.05. ASV relative abundance was visualised using Pheatmap[93], based on Euclidean distance clustering. We also used the function *multipatt* in IndicSpecies[94] to identify ASVs significantly associated with species and location. *Multipatt* uses the function Indval.g[95] to correct for unequal sample sizes and 9 999 permutations to estimate statistical significance. Samples were grouped for the analyses (a) by species considering all the locations, (b) by species only in shared locations (FUN and GUA) and (c) by habitat only in shared locations. Parameters A (specificity) and B (fidelity) were used to assess the predictive value of the ASVs for the location or species, respectively, and their sensitivity as indicators of the group.

Predicted community metagenomic profiling was performed using PICRUSt2 v2.5.2[96]. Briefly, employing HMMER[97], EPA-NG[98] and GAPPA[98], ASVs were aligned with the reference Integrated Microbial Genomes Database[96] and a phylogenetic tree constructed. Hidden state prediction, employing Castor[99], was then used to predict gene family abundance. ASVs with nearest-sequenced taxon index (NSTI)

values > 2 were filtered from the analysis. Metagenome predictions, accounting for 16 S copy number and ASV relative abundance, were then generated. Whole community enzyme classification (EC) number abundances were calculated and subsequently used to infer MetaCyc pathway abundances using MinPath[100]. Differential analysis of predicted functional pathway representation was performed using ALDEx2 v1.30.0[101], using the *glm* tool with a Holm-Bonferroni FWER correction to identify differences between fish species in different locations and between species in sympatry.

## Fluctuating asymmetry

To assess the potential relationship between genetic, epigenetic and microbiome variation and phenotypic variation, we measured fluctuating asymmetry on three morphometric traits potentially related to fitness (area of the caudal ocellus, distance between the eye and the snout and pupil diameter) in 21 fish (11 *K. ocellatus* and 10 *K. hermaphroditus*), all analysed for microbiome and genetic diversity, including those that had epigenetic information as well (Supplementary Data 1). For the ocellus, digital photographs were taken on both sides of the fish against a scale and the area of the ocellus was measured using Image J[102]. Pupil diameter and distance between the tip of the snout and the posterior edge of each eye were measured on preserved specimens using a microscope at 2 x magnification. Two measurements were carried out by the same observer, separated 2 – 4weeks to reduce observer bias.

## Statistical analysis

All statistical analysis were carried out in R v4.2.2[103]. We used linear models to examine the influence of species, sampling location, and fish size on measures of alpha diversity (Shannon diversity, Chao1 richness, Simpson's evenness, and Faith's phylogenetic diversity). Model comparison was carried out by examining changes in AIC using the *anova* command. As sample sizes were relatively small and unequal among sampling sites, we ran non-parametric bootstrapping regressions with 1000 replicates for all linear models, using the *tidymodels* package[104]. Nonparametric bootstrapping involved random sampling with replacement from the dataset to generate a set of new distributions; linear models were run on each one of these datasets.

Structural analysis (microbial beta diversity) was based on community distance matrices calculated using the Bray-Curtis dissimilarity index and the weighted UniFrac distance to take into account phylogenetic relationships among taxa. Non-metric multidimensional scaling ordination was performed using the *vegan* package[105]. To examine the influence of fish species, sampling location and fish length on community structure, multivariate analysis of variance (PERMANOVA) was performed using *adonis*[105] with 99,999 permutations.

To assess the influence of genetic variation on microbiome diversity, individual heterozygosity was calculated in GeneAlex v.6.5.1b[106] from 28 fish (14 *K. ocelatus* and 14 *K. hermaphroditus*) from FUN and GUA locations previously genotyped for 5477 SNPs[51]. We used linear models on to analyse measures of alpha diversity, with heterozygosity, species and sampling location as predictors, controlling for

size differences. The *drop1* function was used to perform variable selection by comparing the full model to reduced models. We compared the full model to each of the reduced models using AIC values and used the likelihood ratio test to compare model fits. We also analysed the relationship between genetic differentiation (pairwise Euclidean genetic distance based on SNP frequencies within individuals) and microbiome dissimilarity (based on Bray-Curtis and weighted Unifract distance) using a Mantel test implemented in the *ecodist* package[107].

Finally, to assess the relationship between the microbiome, genetic and epigenetic patterns we used DNA methylation pairwise distances between individuals previously estimated in[42] for 18 fish (14 *K. hermaphroditus* and 4 *K. ocellatus*) from the sympatric locations (GUA and FUN). We assessed the relationships among the epigenetic pairwise distance (Bray-Curtis), Euclidean genetic distance (based on SNP frequencies within individuals) and microbiome dissimilarity (weighted Unifrac and Bray-Curtis distances) with multiple regression on distance matrices (MRM), using the function MRM in the *ecodist* package. For this analysis we carried out 1000 permutations. In this way, we evaluated all the explanatory variables while accounting for the non-independence of distance matrices.

As a measure of individual epigenetic diversity, we also estimated the coefficient of variation (CV) of the counts per million of 64,152 methylated sites[42] and fitted linear models including alpha diversity, individual heterozygosity, size and species as a predictors. Models were checked for various assumptions using the package *performance*[108] and, after removing collinearity, model selection was carried out using the *drop1* function as above.

For FA analyses we followed the steps recommended in[52] (version updated in 1996). We first tested the assumptions that (a) the difference in size between left and right measurements (L-R; Supplementary Data 8) was not different from zero, using one sample t-tests and (b) that side differences (L-R) were normally distributed, using Shapiro-Wilk's tests. After these tests, only ocellus area and pupil diameter were retained for further analyses, as the snout distance distribution was not normally distributed, indicative of antisymmetry. A linear regression model was then used to test if the absolute difference between left and right trait measurements (|L-R|) was dependent on trait size, and the relationship was found not significant in all cases (Supplementary Fig. 6d). A two-way ANOVA (sides x individuals) was used for testing the significance of FA relative to measurement error, while simultaneously testing for the presence of directional asymmetry (DA) and for trait-size differences among individuals. The results of the significance tests for the various components of variation derived from the two-way ANOVA of the ocellus and pupil measurements indicated that FA was large relative to measurement error as well as a minor contribution of DA (Supplementary Data 9). Corrections for DA and error measurements (between replicates) were carried out as in[109], and these corrected FA estimates were used for the rest of the analyses (Supplementary Data 8). Total fluctuating asymmetry was estimated as the sum of both ocellus and pupil diameter corrected FA, and we assessed its relationship with microbiome (alpha diversity), genetic (heterozygosity) and epigenetic (CV) diversities using linear regressions. Pearson correlation probabilities were tested using Monte-Carlo simulations with 1000 permutations, to account for the small sample sizes.

### Reporting summary
Further information on research design is available in the Nature Portfolio Reporting Summary linked to this article.

## Data availability
Microbiome sequences have been submitted to the European Nucleotide Archive (ENA) under accession number PRJEB61741: FastaQC files for the GBS library for the SNPs and DNA methylation Data can be accessed at NCBI accession number PRJNA563625: Scripts for DNA methylation and SNP bioinformatics processing are available at: https://github.com/waldirmbf/BerbelFilho_etal_KryptolebiasHybridisation/tree/master/1.ProcessingSequencingFiles/1.2.EpigeneticAnalysis[51]. Source Data provided as a source Data file. Source Data are provided with this paper.

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

## Acknowledgements

We are grateful to ICMbio for providing help with accommodation and facilities and to Sergio Lima, Helder M.V. Espírito-Santo, Mateus Lira for support during sample collections. Fieldwork was supported by the National Geographic/Waitt program [W461-16] to SC and by a scholarship from the Conselho Nacional de Desenvolvimento Científico e Tecnológico (CNPq) to WMB-F. I.A. was supported by a Commonwealth PhD Scholarship (BDCS-2020-41). S.C. was partially funded by a Royal Society Industry Fellowship Ref: IF\R1\231030.

## Author contributions

I.A.: lab analysis, bioinformatics, and first draft; T.U.W.: experimental design, bioinformatics and Data analysis, W.B.F.: sample collection, Data analysis; M.H.: sequencing and bioinformatics; B.O. & S.W.: lab and FA analysis; C.G.L.: funding and Data analysis; S.C.: funding, experimental design, Data analysis, and manuscript writing. All authors contributed comments and revisions to the different versions of the manuscript.

## Competing interests

The authors declare no competing interests.
