## [Peer Review File · Nature Communications]

Microbiome and epigenetic variation in wild fish with low genetic diversityREVIEWER COMMENTS

Reviewer #1 (Remarks to the Author):

Dear Authors,

I have now reviewed MS NCOMMS-23-20774. My feeling is that this is a good field (and comparative) study describing interesting patterns of covariation among different diversity facets (microbiome, epigenetic and genetic). However, it does not provide insights into the role of these facets of diversity for phenotypic variation or phenotypic plasticity, as claimed by the authors in the title and the abstract. Although some of the patterns reported here are interesting, because of its correlative nature (and the relatively low sample sizes per species and per site), its mechanistic relevance is limited, which in my opinion reduces its interest for a broad readership.

Overall, I found the MS clear and well-written, but I have several comments more directly related to the methods, results and the interpretation.

-First, as I told before, this is a descriptive and correlative study, without any mechanistic insights onto (i) the processes generating these patterns and (ii) the consequences for phenotypic variation and fitness. The title and the abstract (especially the last sentence) have to be modified to better reflect the content of the MS.

-The sample sizes vary among sites and species by a factor 3. I have not seen in the M&M any procedures allowing to take this variability in sample size into account for the estimates of diversity. It exists some rarefaction procedures (for microbial diversity but also for genetic and epigenetic diversity) that I have not read mentioned or cited in the text. If authors did not take difference in sample size into account, they should as this has strong influence on estimates of alpha diversity.

-If I have understood correctly the previous papers, epigenetic diversity profiles have been taken from fins, whereas the microbiome has been taken from the skin mucus. Epigenetic profiles are known to be tissue-dependent; authors suggest that the epigenetic diversity of the fins can be influenced by the microbiome of the skin mucus. How is it possible? In my opinion, authors are reporting co-variation, not causal links. They should be more careful all over the MS in their interpretation, many of them are probably not causal, but just reflect that diversity patterns co-vary. This is quite well known in ecology, see for instance the papers by Vellend (Vellend, *Am Nat* 2005; Vellend and Geber, *Ecol Lett* 2005) at other scales: diversity patterns are governed by similar processes and are hence expected to co-vary. This is particularly obvious for genetic and epigenetic diversity (Herrera et al. *Mol Ecol* 2016; Fargeot et al. *Genes* 2021) as the later is often controlled by the former. They should revisit most of their interpretation as co-varying patterns rather than as causes and consequences.

-I have a problem with linear models used to test patterns of differentiation (the effects of genetic and microbiome differentiation on epigenetic differentiation). You are dealing with matrices of dissimilarity, which poses problems of pseudo-replication. Mantel tests (based on permutations) are generally used to deal with this problem. Linear mixed models can also be used, but not as you formulate your models (including species and location twice in the model is a total non sense, I understand why you do that but this does not take the problem into account, the way you formulated the random terms neither). Mantel tests are limited to two variables, but it exists extension of the Mantel test for more than two variables, and especially "regressions on distance matrices" (function MRM, R package ecodist) that would better fit the type of data you are modeling.

-There are some parts in the discussion that you should revisit, as you have no mechanistic cues about what is going on. For instance: l 278: "our results indicate that the skin microbiome can also influence the host epigenetics". No this co-variation is perhaps simply due to the fact that they are both controlled by similar environmental factors, which lead to a similar pattern.

Reviewer #2 (Remarks to the Author):

The paper by Anka et al. reports metabarcoding of the bacterial component of the skin microbiome in two closely related species of mangrove killifish, one an outcrossing species, the other a self-crossing species. The sampling design is robust (6 populations, two of each type) and the analyses of the 16S metabarcode data is appropriate and addresses important, and still controversial, questions about the role of intrinsic vs extrinsic factors affecting organisms' microbiome. The inclusion of the two life histories in those analyses makes this work even more of interest to a broad readership. The authors then use previously collected (and published) data (DNA methylation and genetic diversity (SNPs)) for the same populations to test for relationships between host genetic diversity, epigenome and microbiome. While the previously collected data was limited to subsets of the microbiome sampling, it still allowed a powerful and interesting analysis of these complex relationships – making this work unique, at least to my knowledge. There are limitations to this dataset, and I have some comments and questions (see below), but overall, I think this work is important, interesting and relevant to the emerging field of variation in non-genetically coded plasticity in natural populations.

Comments/Questions

1) The sample sizes used for microbiome analyses are small, although I think they are sufficient to support the bacterial community analyses described here. Sample size limitations become a bit more worrisome when the previously collected data is included, as those data are for a sub-sample of the microbiome sampled fish (note: please make it clear that metabarcoding, methylation analysis and SNP genotyping were performed on the same fish). I suggest the authors include some form of power analysis to support their conclusion, despite the limited sample sizes for some species/locations.

2) I believe a truly unique component of this study is the inclusion of the two life histories – quite apart from the resulting different levels of inbreeding/genetic diversity. I suggest including more explicit reference to not only “species” effects, but the fact that species is also life history. It is also important to test (and report) explicitly for sympatric versus non-sympatric population effects.

3) Technical: Bray-Curtis and UniFrac analyses (adonis) is usually referred to as PERMANOVA, perhaps make that clear? Additionally, the authors should consider analyzing principle coordinates from a B-C or UniFrac matrix PCoA analysis to test for location/species etc. effects. This may be redundant though.

4) Given that one species has recently colonized the mixed populations, is there any possible hybridization in mixed communities? I don't think *K. hermaphroditus* are obligatory self-crossing? It might be valuable to test for gene flow between the species in the mixed populations. This is not a flaw, just perhaps an interesting twist.

5) The relationship between the host microbiome and epigenetics is complicated – which drives which? Only host genomic diversity (SNP genotypes) is fixed and can reliably be set as an independent variable. I do understand that the authors can make an argument for the microbiome to most likely affect host methylation (rather than the reverse), but in fact both are possible (and both may happen). I suggest the authors consider this in their Discussion.

6), The authors comment on genotype-by-environment effects, and I assume they are referring to their significant species-by-location effects; however, those effects are likely due to a lot of different mechanisms, and the authors need to be cautious about interpreting species-by-location effects as being GxE. Similarly, the authors need to make a better case for their results supporting broadly defined epigenetic (DNA methylation and the skin microbiome) effects on adaptive plasticity.

Daniel D. Heath

RESPONSE TO REVIEWERS' COMMENTS

In this revised version we have addressed all comments suggested by the two reviewers. The main changes are as follows:

1. Following the suggestions of the reviewers, we have now carried out additional statistical analyses based on resampling (bootstrapping and permutations), to address the relatively small sample sizes and the difference among locations.
2. We have added new analyses on fluctuating asymmetry of three phenotypic traits, potentially related to fish performance, and assessed their relationship to genetic, epigenetic and microbiome diversity.
3. We have also carried out new multivariate regression analyses and mantel tests for testing the association between microbiome, genetic and epigenetic distance matrices.
4. We have expanded the analyses and discussion related to the sympatric/allopatric status of the populations.
5. We have expanded the consideration to the different mating systems across the discussion.
6. We have modified the title, abstract and discussion to reflect the field nature of our study and clarified that from our results we cannot infer causality throughout. However, we can offer a plausible adaptive mechanism that can inform further studies, using a unique study system. While field studies can rarely infer causality, the study of allopatric and sympatric populations can be a powerful way of examining the effects of natural selection in the wild ^{1, 2}.

Below we include a detailed response to all the comments.

REVIEWERS' COMMENTS

Reviewer #1 (Remarks to the Author):

Dear Authors,

- I have now reviewed MS NCOMMS-23-20774. My feeling is that this is a good field (and comparative) study describing interesting patterns of covariation among different diversity facets (microbiome, epigenetic and genetic). However, it does not provide insights into the role of these facets of diversity for phenotypic variation or phenotypic plasticity, as claimed by the authors in the title and the abstract. Although some of the patterns reported here are interesting, because of its correlative nature (and the relatively low sample sizes per species and per site), its mechanistic relevance is limited, which in my opinion reduces its interest for a broad readership.

We agree with the reviewer that the nature of our field study does not allow us to infer the mechanisms underlying the associations we observed among the different variables, we have made this clear in the manuscript (lines 322-325; 351-354; 104-06), removed the discussion relating these associations to phenotypic plasticity and reworded the discussion throughout to refer to associations between microbiome and epigenome, instead of

interactions (see document with track changes). We have also modified the title and abstract to reflect this. Additionally, we have included new analyses of fluctuating asymmetry in phenotypic traits (lines 223-237, 325-343, 438-448) to try inferring some of the potential consequences of the variation we observed. We believe that this has strengthened the manuscript and that the results are of interest for a broad readership, particularly for researchers working on the role of the microbiome and epigenetics in evolutionary ecology and conservation biology. We also note that although field studies like ours cannot infer causality, the study of allopatric and sympatric populations can be a powerful way of examining the effects of natural selection in the wild ^{1,2}

Overall, I found the MS clear and well-written, but I have several comments more directly related to the methods, results and the interpretation.

-First, as I told before, this is a descriptive and correlative study, without any mechanistic insights onto (i) the processes generating these patterns and (ii) the consequences for phenotypic variation and fitness. The title and the abstract (especially the last sentence) have to be modified to better reflect the content of the MS.

As discussed above, we agree with the reviewer that due to the nature of the study (field-based), its mechanistic insights are necessarily limited. As suggested, we have modified the title, abstract and discussion to reflect this (lines 322-325, 345-356).

-The sample sizes vary among sites and species by a factor 3. I have not seen in the M&M any procedures allowing to take this variability in sample size into account for the estimates of diversity. It exists some rarefaction procedures (for microbial diversity but also for genetic and epigenetic diversity) that I have not read mentioned or cited in the text. If authors did not take difference in sample size into account, they should as this has strong influence on estimates of alpha diversity.

Thank you for this comment. We appreciate that, although the estimates of microbial (alpha-diversity), genetic (H_e) and epigenetic (CV) diversity were individually and not population-based estimated, the differences in sample sizes could have affected the results of the statistical analyses. To account for this potential effect, we have carried out additional non-parametric bootstrapped regression analyses on the alpha diversity models, using 1000 resamplings. The results from these analyses supported our previous results in all cases and we have now described this in methods (lines 455-458), results (157-158, 198-199) and supplementary materials (Table S6), including figures comparing the bootstrapped and original distributions (Figure S2). Regression analyses have also been carried out using a Monte-Carlo approach to account for unequal sample sizes (lines 210-217, 232-236, 474-481, 502-507). We believe that this approach has strengthen our results.

-If I have understood correctly the previous papers, epigenetic diversity profiles have been taken from fins, whereas the microbiome has been taken from the skin mucus. Epigenetic profiles are known to be tissue-dependent; authors suggests that the epigenetic diversity of the fins can be influenced by the microbiome of the skin mucus. How is it possible?

Yes, this is correct, the skin microbiome was taken with a swab of the whole flank of the fish. As the mucus covers all the surface of the fish, including the fins, and provided the small size of the fish, we trust that any relationship between the skin microbiome and the epigenome will include the fins as well as the rest of the epidermis. The fish epidermal

mucus plays a key role in the innate immunity of the fish and skin, including fins, represents one of the main routes of infection, therefore we expect a close link between the external microbiota and skin/fin cells.

-In my opinion, authors are reporting co-variation, not causal links. They should be more careful all over the MS in their interpretation, many of them are probably not causal, but just reflect that diversity patterns co-vary. This is quite well known in ecology, see for instance the papers by Vellend (Vellend, *Am Nat* 2005; Vellend and Geber, *Ecol Lett* 2005) at other scales: diversity patterns are governed by similar processes and are hence expected to co-vary. This is particularly obvious for genetic and epigenetic diversity (Herrera et al. *Mol Ecol* 2016; Fargeot et al. *Genes* 2021) as the later is often controlled by the former. They should revisit most of their interpretation as co-varying patterns rather than as causes and consequences.

The first part of this comment has been responded above. We totally agree with the reviewer that at least part of the epigenetic variation is determined by the underlying genetic variation, we have now made this clearer and added the corresponding references (lines 303-306). We appreciated the suggestion of the references regarding the correlation between species and generic diversity, which we agree could apply to part of our results. We have now included this interpretation and the relevant references in the discussion (lines 281-284, 350-353). We believe that this deeper interpretation of the correlation patterns has made clearer the relevance of the part of the epigenetic and microbiome variation unrelated to the underlying genetic diversity (lines 308-319).

-I have a problem with linear models used to test patterns of differentiation (the effects of genetic and microbiome differentiation on epigenetic differentiation). You are dealing with matrices of dissimilarity, which posits problems of pseudo-replication. Mantel tests (based on permutations) are generally used to deal with this problem. Linear mixed models can also be used, but not as you formulate you models (including species and location twice in the model is a total non sense, I understand why you do that but this does not take the problem into account, the way you formulated the random terms neither). Mantel tests are limited to two variables, but it exists extension of the Mantel test for more than two variables, and especially "regressions on distance matrices" (function MRM, R package *ecodist*) that would better fit the type of data you are modeling.

We very much appreciate this comment of the reviewer and have now run the suggested analyses, both Mantel and MRM tests, using the *ecodist* package (lines 198-204, 210-217, 232-236, 474-481). As for the previous analyses, we found that these confirmed and strengthened our results.

-There are some parts in the discussion that you should revisit, as you have no mechanistic cues about what is going on. For instance: l 278: "our results indicate that the skin microbiome can also influence the host epigenetics". No this co-variation is perhaps simply due to the fact that they are both controlled by similar environmental factors, which lead to a similar pattern.

As mentioned in the responses above, we have extensively modified our discussion to clarify the nature of the observed associations (see track changes across the ms) and included the consideration to potential co-variation of the different variables analysed (lines 344-355).

Reviewer #2 (Remarks to the Author):

The paper by Anka et al. reports metabarcoding of the bacterial component of the skin microbiome in two closely related species of mangrove killifish, one an outcrossing species, the other a self-crossing species. The sampling design is robust (6 populations, two of each type) and the analyses of the 16S metabarcode data is appropriate and addresses important, and still controversial, questions about the role of intrinsic vs extrinsic factors affecting organisms' microbiome. The inclusion of the two life histories in those analyses makes this work even more of interest to a broad readership. The authors then use previously collected (and published) data (DNA methylation and genetic diversity (SNPs)) for the same populations to test for relationships between host genetic diversity, epigenome and microbiome. While the previously collected data was limited to subsets of the microbiome sampling, it still allowed a powerful and interesting analysis of these complex relationships – making this work unique, at least to my knowledge. There are limitations to this dataset, and I have some comments and questions (see below), but overall, I think this work is important, interesting and relevant to the emerging field of variation in non-genetically coded plasticity in natural populations.

We appreciate the positive comments of the reviewer with regards to the uniqueness of the study, the appropriateness of the experimental design and relevance of the results. We acknowledge that the data set has limitations and have tried to overcome these by following the suggestions of both reviewers and adding new statistical analyses and measurements. The details of this new analyses are expanded in the response to both reviewers.

Comments/Questions

1) The sample sizes used for microbiome analyses are small, although I think they are sufficient to support the bacterial community analyses described here. Sample size limitations become a bit more worrisome when the previously collected data is included, as those data are for a sub-sample of the microbiome sampled fish (note: please make it clear that metabarcoding, methylation analysis and SNP genotyping were performed on the same fish). I suggest the authors include some form of power analysis to support their conclusion, despite the limited sample sizes for some species/locations.

We have now further clarified in the different analyses that when the different variables jointly analysed (i.e., genetic diversity, epigenetic diversity, microbiome diversity) correspond to the same fish (hence the variable sample sizes of some of the datasets) (lines 440-443, 474-476). To account for the relatively small sample sizes and unequal distribution among locations, we have now run non-parametric analyses using bootstrapping and permutations, to test the robustness of our results. The new analyses (lines 157-158, 198-204 & 210-217, 232-236, 474-481, 502-507) supported and strengthened our previous results, and more details are provided in the response to the R1 above.

2) I believe a truly unique component of this study is the inclusion of the two life histories – quite apart from the resulting different levels of inbreeding/genetic diversity. I suggest including more explicit reference to not only “species” effects, but the fact that species is also life history. It is also important to test (and report) explicitly for sympatric versus non-sympatric population effects.

We appreciate this comment and have now added new analyses specifically referring to the sympatric versus non-sympatric status of the populations (lines 98-106, 169-176, 184-189; Figures S3 & S4), together with more information on the potential role of their respective life histories in the patterns observed in the discussion (lines 183-184, 278-281, 335-342). We believe this approach has further clarified the roles of species and environment on the results.

3) Technical: Bray-Curtis and UniFrac analyses (adonis) is usually referred to as PERMANOVA, perhaps make that clear? Additionally, the authors should consider analyzing principle coordinates from a B-C or UniFrac matrix PCoA analysis to test for location/species etc. effects. This may be redundant though.

We have clarified the permanova term (line 462) and, following the recommendations of both reviewers, we have performed new multivariate analysis among the distance matrices. As the reviewer indicates, the PCoA would be largely redundant to the current analyses and the NMDS visualisation so, in order to keep the ms within word limits, we have opted for not adding this analysis.

4) Given that one species has recently colonized the mixed populations, is there any possible hybridization in mixed communities? I don't think *K. hermaphroditus* are obligatory self-crossing? It might be valuable to test for gene flow between the species in the mixed populations. This is not a flaw, just perhaps an interesting twist.

Indeed, we found evidence of hybridisation between both species in a previous study, despite their different mating systems. We have added this aspect to the discussion and found that it is an important factor in explaining the relationship between fluctuating asymmetry and heterozygosity (lines 335-342).

5) The relationship between the host microbiome and epigenetics is complicated – which drives which? Only host genomic diversity (SNP genotypes) is fixed and can reliably be set as an independent variable. I do understand that the authors can make an argument for the microbiome to most likely affect host methylation (rather than the reverse), but in fact both are possible (and both may happen). I suggest the authors consider this in their Discussion. We acknowledge this is a possibility and have now considered this in the discussion (lines 52-53, 241-244, 350-353).

6), The authors comment on genotype-by-environment effects, and I assume they are referring to their significant species-by-location effects; however, those effects are likely due to a lot of different mechanisms, and the authors need to be cautious about interpreting species-by-location effects as being GxE. Similarly, the authors need to make a better case for their results supporting broadly defined epigenetic (DNA methylation and the skin microbiome) effects on adaptive plasticity.

We agree with both of these comments, accordingly we have removed the discussion regarding phenotypic plasticity and the reference to genotype by environment interactions, instead we have discussed the results in relation to the new analyses of fluctuating asymmetry and the roles of genetic diversity and environment, without assuming a GxE interaction (lines 9-10, 344-345).

References

1. Endler JA. *Natural selection in the wild*. Princeton University Press (1986).
2. Mousseau TA, Sinervo B, Endler JA. *Adaptive genetic variation in the wild*. Oxford University Press (2000).

REVIEWER COMMENTS

Reviewer #1 (Remarks to the Author):

Dear Authors,

The MS has been much improved, and as far as I can judge, most remarks have been carefully addressed. My personal feeling is still that it is a good empirical study (we need this type of study) that is yet limited by the sample sizes and by its correlative nature.

I still have a series of comments that I list below:

- I may be wrong, but paragraph starting l. 169, p. 9 ("Chao1 diversity was significantly...") is redundant with paragraph starting l. 143 in which you already describe these diversity indices
- Regarding the model described l. 190: you remove "species" from the model because of collinearity, which is fine. You find an effect of heterozygosity and the plot you show (fig 5) display a tendency for each species. This is not what you have modeled as you have removed species from the model. If I understood your model properly you should have a single equation linking (eg) Chao1 to heterozygosity, whereas here you display two equations (except if you've done one model per species). Please clarify this by providing more details. Moreover, as you remove "species" from the model, and as species and heterozygosity are partially confounded, this is perhaps the "species" effect that you detect here. Perhaps you may consider a mixed model with species as the random term.
- You use an Euclidian distance of genetic diversity. I don't understand how an Euclidian distance can be estimated from SNP data. Euclidian distance is a difference between two quantitative entities. I do not figure out a distance can be measured between two categorical SNPs. There are plenty of dissimilarities that can be estimated from genetic data (F_{st} or even Bray-Curtis) and it is quite unusual to see an Euclidian distance. Please clarify or justify
- In the Discussion, the paragraph of hybridization is not convincing and I think this is not the most parsimonious hypothesis to explain the patterns. Perhaps outbreeding depression is a better explanation.
- Figures 6 & 7 are really poor. Please improve the quality of these figures.

Reviewer #2 (Remarks to the Author):

The authors have addressed all of my substantive concerns. I like the inclusion of additional statistical analyses and the bootstrapping to address sample size concerns. The inclusion of additional considerations associated with the life histories of the two species, the potential for a history of hybridization in the sympatric populations and the differences between associations and causation are clear and important. The authors chose to include fluctuating asymmetry (FA) data to their study, and while they note early on that there is considerable controversy about the fitness implications of FA, they seem to still use their FA data as evidence for fitness effects. I suggest caution on that; however, I do believe the inclusion of the FA data adds to this already interesting study. There are a few minor grammatical issues (particularly in the new text), and perhaps the MS should be proof-read. Overall, a very strong paper dealing with an important topic.

RESPONSE TO REVIEWERS' COMMENTS

In this revised version we have addressed all the queries of the two reviewers (see detailed responses below) and highlighted the changes made in the main document.

REVIEWER COMMENTS

Reviewer #1 (Remarks to the Author):

Dear Authors,

The MS has been much improved, and as far as I can judge, most remarks have been carefully addressed. My personal feeling is still that it is a good empirical study (we need this type of study) that is yet limited by the sample sizes and by its correlative nature.

I still have a series of comments that I list below:

-I may be wrong, but paragraph starting l. 169, p. 9 ("Chao1 diversity was significantly...") is redundant with paragraph starting l. 143 in which you already describe these diversity indices

Although these paragraphs read similar, they correspond to different analyses and have different results. The first analysis assessed the influence of species, sampling location and fish size on measures of alpha diversity while the second one assessed the sympatric versus allopatric status of the species (instead of the particular location). We have added a sentence at the start of line 170 to clarify the difference (i.e., *When we considered the occurrence of both species in sympatry or allopatry...*).

-Regarding the model described l. 190: you remove "species" from the model because of collinearity, which is fine. You find an effect of heterozygosity and the plot you show (fig 5) display a tendency for each species. This is not what you have modeled as you have removed species from the model. If I understood your model properly you should have a single equation linking (eg) Chao1 to heterozygosity, whereas here you display two equations (except if you've done one model per species). Please clarify this by providing more details. Moreover, as you remove "species" from the model, and as species and heterozygosity are partially confounded, this is perhaps the "species" effect that you detect here. Perhaps you may consider a mixed model with species as the random term.

Yes, we removed species due to collinearity with heterozygosity and in Figure 5 we were showing that the trend was the same for heterozygosity and alpha diversity in both species, so the relationship between heterozygosity and alpha diversity was not due to the species, despite their differences in the level of heterozygosity. However, we realise that the figure displaying species separate can be confusing and we have now replaced it by a new Figure 5 without the species separation. Using species as a random effect with only two levels would not be appropriate as at least 4 or 5 levels are generally recommended¹. Although, there has been some recent discussion about the possibilities of using random effects with less than five levels, it is still not recommended for low sample sizes², as in this case.

-You use an Euclidian distance of genetic diversity. I don't understand how an Euclidian distance can be estimated from SNP data. Euclidian distance is a difference between two quantitative entities. I do not figure out a distance can be measured between two

categorical SNPs. There are plenty of dissimilarities that can be estimated from genetic data (Fst or even Bray-Curtis) and it is quite unusual to see an Euclidian distance. Please clarify or justify

We have clarified that the Euclidean distance was estimated based on allele frequency within individuals (lines 474 & 480). Euclidean distance is often the distance of choice in SNP studies for both individuals and populations^{3, 4, 5}. In contrast, Bray-Curtis is not recommended for SNPs as it uses the abundance of the alternate allele, and hence is which renders it asymmetric because the choice of reference/alternate allele is arbitrary⁶.

-In the Discussion, the paragraph of hybridization is not convincing and I think this is not the most parsimonious hypothesis to explain the patterns. Perhaps outbreeding depression is a better explanation.

We have now included outbreeding depression as a potential additional explanation, including a relevant reference (lines 339-342).

-Figures 6 & 7 are really poor. Please improve the quality of these figures.

We have improved the quality of these images and we are also providing independent TIFF files for each one of these.

Reviewer #2 (Remarks to the Author):

The authors have addressed all of my substantive concerns. I like the inclusion of additional statistical analyses and the bootstrapping to address sample size concerns. The inclusion of additional considerations associated with the life histories of the two species, the potential for a history of hybridization in the sympatric populations and the differences between associations and causation are clear and important. The authors chose to include fluctuating asymmetry (FA) data to their study, and while they note early on that there is considerable controversy about the fitness implications of FA, they seem to still use their FA data as evidence for fitness effects. I suggest caution on that; however, I do believe the inclusion of the FA data adds to this already interesting study. There are a few minor grammatical issues (particularly in the new text), and perhaps the MS should be proof-read. Overall, a very strong paper dealing with an important topic.

Many thanks for the positive comments, we have now proof-read the whole text and corrected the grammatical issues (see document with track changes).

References

1. Arnqvist G. Mixed models offer no freedom from degrees of freedom. *Trends Ecol Evol* **35**, 329-335 (2020).
2. Gomes DG. Should I use fixed effects or random effects when I have fewer than five levels of a grouping factor in a mixed-effects model? *PeerJ* **10**, e12794 (2022).

3. Ghaffari S, Hasnaoui N, Zinelabidine LH, Ferchichi A, Martínez-Zapater JM, Ibáñez J. Genetic diversity and parentage of Tunisian wild and cultivated grapevines (*Vitis vinifera* L.) as revealed by single nucleotide polymorphism (SNP) markers. *Tree genetics & genomes* **10**, 1103-1112 (2014).
4. Ketema S, *et al.* DArTSeq SNP-based markers revealed high genetic diversity and structured population in Ethiopian cowpea [*Vigna unguiculata* (L.) Walp] germplasms. *PloS one* **15**, e0239122 (2020).
5. Hayah I, Talbi C, Chafai N, Houaga I, Badaoui B. Genetic diversity and breed-informative SNPs identification in domestic pig populations using coding SNPs. *Frontiers in genetics* **14**, 1229741 (2023).
6. Georges A, Mijangos L, Kilian A, Patel H, Aitkens M, Gruber B. Distances and their visualization in studies of spatial-temporal genetic variation using single nucleotide polymorphisms (SNPs). *bioRxiv*, 2023.2003. 2022.533737 (2023).